# Predicting Ego Integrity Using Prior Ego Development Stages for Older Adults in the Community

**DOI:** 10.3390/ijerph18189490

**Published:** 2021-09-08

**Authors:** Pei-Yun Chen, Wen-Chao Ho, Chyi Lo, Tzu-Pei Yeh

**Affiliations:** 1Department of Public Health, China Medical University, Taichung 406040, Taiwan; peiyun0203@gmail.com (P.-Y.C.); wcho@mail.cmu.edu.tw (W.-C.H.); 2School of Nursing, China Medical University, Taichung 406040, Taiwan; 3Department of Nursing, China Medical University Hospital, Taichung 404332, Taiwan

**Keywords:** personality, successful aging, death, life course and developmental change

## Abstract

Background: Erikson’s ego development theory is the most accepted theory that involves eight stages of psychosocial development over an individual`s all lifespan. The result of development in prior stages will influence the later stages. The elderly were mainly characterized by the central developmental tasks: achieving ego integrity vs. despair. The harvest in the last stage will be related to the attitude of facing death in the elderly. Methods: A cross-sectional study of elderly age from 65 to 90 years old (*n* = 292) was carried out and investigated via the Inventory of Psychosocial Balance. Pearson correlation and path analysis were performed in order to analyze the direct and indirect effect among the first seven stages with the eighth stage. Results: We found that all the eight stages were significantly related to each other, and comparing to the previous seven stages, “the generativity stage” (r = 0.77) was the most relevant stage with “ego integrity”. In all indirect and direct effects, the seventh stage had the greatest impact on the “ego integrity stage”; the direct effect was 0.89. Conclusions: Compared to the whole lifespan, adulthood possessed a higher influence on the elderly stage. We found that all the eight stages were significantly related to each other, and comparing the first seven stages, the “generativity stage” (r = 0.77) was the most relevant stage to “ego integrity”. Conclusions: Compared to the whole lifespan, adulthood possessed a higher influence on the elderly stage.

## 1. Introduction

Caring for older adults has become an increasingly important global issue as life expectancy increases. Although advances in medical science have helped people survive and cope with chronic diseases, the well-being of older adults concerns their mental health in addition to their physical health. Older adults experience a unique set of challenges, including physical decline [1]; cognitive decline; and psychosocial changes, such as bereavement, loneliness, and depression [1,2]. Inevitably, death is an important issue that older adults near the end of their life must face.

In Taiwan in 2018, 2,060,754 older adults (older than 65 years) visited a psychiatric outpatient department and were diagnosed as having “nonpsychotic mental disorders”, representing a notable increase from the 1,850,146 older adults who received the same diagnosis in 2016. Thus, the psychosocial well-being of older adults is a growing concern. Erikson’s ego development theory is a robust theory of psychosocial development over an individual’s lifespan. Erikson’s theory characterizes older adults as facing the central developmental task of achieving ego integrity in opposition to despair. The theory also conceives wisdom and well-being as products of successful human development, where early childhood interventions may plant the “seeds of wisdom” that bear fruit later in life [3,4].

Theoretically, the psychosocial development of older adults is related to their past life experiences. Studies have mainly adhered to the Eriksonian tradition, where ego integrity and despair are related to the resolution of earlier psychosocial crises, acceptance of the past, and finitude of life [5]. However, few studies have investigated how the first seven stages, which culminate in ego integrity, can be predicted. Consequently, it remains uncertain how the first seven stages affect later stages. Thus, this research aimed to (a) investigate the relationships among the first seven stages and between the first seven stages and ego integrity in older adults, (b) explore the direct and indirect effects of the eight stages of ego development, and (c) formulate a predictive model in which the first seven stages are used to predict ego integrity.

### 1.1. Erikson’s Ego Development Theory

According to Erikson’s theory, older adults are in the last stage of ego integrity and despair, which constitute one of the fascinating conceptual pairs that Erikson coined in his theory of ego development across the life span [5]. Older adults have already experienced most of their idiographic life, and they might thus begin reflecting on how they have lived their life. If an older adult recognizes themselves as having lived a worthy life with no regrets, they face death calmly, which is reflected in ego integrity [6]. Erikson [7] described ego integrity as “the acceptance of one’s one and only life cycle as something that had to be”. People in the final stage of life must come to deal with their past and must find meaning in their lives to face their death [8]. This involves dealing with regrets, feelings of hatred, and past conflicts that they ought to resolve [9]. If a person cannot accept their past, then they fail to achieve ego integrity and tend toward ego despair. If an older adult in later life has reasons for despair, then aspects of the individual’s past, present, and future cannot be easily integrated into a meaningful whole [5,10].

According to Erikson’s ego development theory, people experience eight stages of ego development in their lifespan from infancy to late older adulthood; from earliest to latest, these are “trust versus mistrust (infancy)”, “autonomy versus shame (early childhood)”, “initiative versus guilt (preschool age)”, “industry versus inferiority (school age)”, “identity versus role confusion (adolescence)”, “intimacy versus isolation (early adulthood)”, “generativity versus stagnation (adulthood)”, and “integrity versus despair (late adulthood, older adult)” [11]. Each stage has a distinctive task and conflict [7]. The first stage is “trust versus mistrust”. In this stage, infants depend on their caregiver(s) for survival. If the infant’s needs are not satisfied, then they are likely to become mistrustful, suspicious, and anxious; success in this stage results in the individual cultivating the virtue of hope [6]. The second stage is “autonomy versus shame”. Individuals in this stage are focused on developing a sense of skillful control over their body and a sense of independence. If the environment is especially harsh or constrained, then the individual is likely to develop low self-esteem and a lack of confidence in their ability to survive [6]. The third stage is “initiative versus guilt”. Along with beginning to interact with other children at school, children in the stage are curious, competitive, and even nosy [7]. If the child in this stage is overly repressed or harshly castigated when they express themselves freely, they are likely to become uncurious and unenthusiastic in their learning. The fourth stage is “industry versus inferiority”. At this stage, the child enters into more complicated interpersonal relationships and starts competing with their peers. If the child at this stage experiences repeated failure, they are likely to develop a sense of inferiority [6].

The fifth stage is “identity versus role confusion”. This is the phase of adolescence, straddling childhood and adulthood, in which the crucial developmental task of negotiating one’s identity (i.e., asking “who am I?”) takes center stage [7,12]. Erikson stated that the adolescent may feel uncomfortable about their changing body for some time before they “grow into” these changes [6]. The sixth stage is “intimacy versus isolation”. During this stage, people are focused on committing themselves to concrete affiliations, most often in the form of romantic relationships [12]. The seventh stage is “generativity versus stagnation”. Erikson defined generativity as “the concern in establishing and guiding the next generation” [6]. At this stage, the adult is concerned with leaving a lasting legacy by cultivating the next generation [6,8]. The last stage is “integrity versus despair”. At this stage, individuals review their lives, either with satisfaction or regret [12]. If an older adult can accept and integrate their past with their present, they tend toward ego integrity. In general, to face death, older adults must come to deal with their past and find meaning in their lives [7].

If a crisis in a prior stage remains unresolved, then the individual may slip into crisis at later stages. Thus, how an individual lives the final stage of ego integrity versus despair depends on how earlier crises are resolved [7].

### 1.2. The Present Study’s Approach in Relation to Erikson’s Ego Development

Erikson’s ego development theory posits that ego development is influenced by life experience. Satisfaction with past life, present life, future life, and depression was significantly correlated with ego integrity and despair in late adulthood [13]. Forgiveness is an important element for the developmental task of integrity and despair. Higher levels of forgiveness were associated with higher levels of integrity and lower levels of despair [14]. In particular, the wisdom of older adults is associated with their experience, specifically of their psychosocial growth facilitated by support during childhood, competence during adolescence, emotional stability in early adulthood, and generativity at midlife [4].

Midlife is a crucial period for ego development in the transition to old age [15]. Middle-aged adults may be responsible for caring for their children or older adult parents, maintaining their marriage, developing a successful career, and leaving a legacy. They are also prone to facing various midlife challenges, such as divorce, the loss of a parent, empty nest syndrome, deteriorating health, and career transitions, which can challenge their belief system and expectations of what life brings [15]. Although adults at midlife are prone to experiencing disruption, these disruptions serve as transformative experiences that allow them to reevaluate their lives and embark on new challenges [16]. A study applied the “Inventory of Psychosocial Balance” to measure the ego development of 520 participants aged 55–84. The results indicated that the eight stages were related to each other. Specifically, relative to the coefficient of the correlation between the ego integrity stage and the first seven stages, the seventh stage was most highly related to the ego integrity stage (r = 0.86), followed by the first stage (r = 0.83). The second stage was the least correlated (r = 0.72) with ego integrity (Hannah et al., 1996). A longitudinal study of the 1951–1996 period based on Erikson’s ego development theory also reported that the seventh stage is the one that is most consequential and correlated (r = 0.49, *p* < 0.05) with ego integrity [17].

Another study used the first seven stages to predict ego integrity through regression analysis. In the regression model, the coefficient of determination (R^2^) of the seventh stage was 0.77. However, the R^2^ of the first seven stages was 0.84, and the seventh stage already contributed 0.77 of the 0.84. This result means that the seventh stage is the most consequential predictor of an individual’s ego integrity [18].

Ego development remains rarely investigated over the past few decades, especially in East Asian contexts.

## 2. Methods

### 2.1. Participants and Sampling Method

This study was approved by an institutional review board (Approval No. DMR95-IRB-144). Cross-sectional study design with questionnaire survey and convenient sampling method were used; the participants included elder adults who were older than 65 years old. The participants were recruited from Central Taiwan at community parks, temples, religious festivals, and in community activity centers. Inclusion criteria included: residents who were living in the local community and older than 65 years old. Exclusion criteria included: having cognitive impairment, living in a long-term care institution, or being unwilling to participate. Participants were approached by a researcher who explained the study purpose and content, and participants signed informed consent before data collection. The Mini-Mental State Examination (MMSE) was used to screen if the participants had cognitive impairment; MMSE scores under 23 are represented as the cut-off score for cognitive impairment, and those participants who had MMSE scores below 23 were excluded in data analysis [19].

G power software was used for sample size estimation by using Linear multiple regression with seven dependent variables and R^2^ as 0.13, which means moderate effects could be used [20]; in order to reach the detection of an indirect effect with 0.8 and set up α error = 0.05, this study should recruited 104 participants. Considering 20% loss rate, 115 participants should be invited. However, since this study analyzed the collected data in path analysis with structural equation modeling (SEM), the sample size should reach 250–500 [21]. Therefore, this study finally aimed to recruit 500 participants. During 1 year period of data collection, 300 participants attended this study, but 8 participants did not complete the whole questionnaires.

### 2.2. Instruments and Research Variables

This study’s questionnaire comprised questions on sociodemographic characteristics (age, gender, religion, and occupation) and questions based on the Inventory of Psychosocial Balance (IPB), which measures how successfully an individual has achieved psychosocial developmental tasks.

Specifically, the IPB was formulated by Domino and Affonso [22] on the basis of Erikson’s theory of the ego development stages. The IPB was composed of eight subscales: “trust versus mistrust (Stage 1)”, “autonomy versus shame (Stage 2)”, “initiative versus guilt (Stage 3)”, “industry versus inferiority (Stage 4)”, “identity versus role confusion (Stage 5)”, “intimacy versus isolation (Stage 6)”, “generativity versus stagnation (Stage 7)”, and “integrity versus despair (Stage 8)”. Each subscale contains 15 items scored on a 5-point Likert scale from strongly disagree (1 point) to strongly agree (5 points); higher scores indicate better ego development. The questionnaire had good test–retest reliability for the eight stages, and the Cronbach’s alpha of the subscales ranged from 0.64 to 0.79 [22].

In this study, the eighth stage, “integrity versus despair (Stage 8)”, represented the ego development of elderly period and was recognized as the dependent variable in this study. The first seven stages were defined as the factors which might influence the eighth stage with certain effects.

### 2.3. Model Construction

In this research, path analysis was used for statistical analysis. The purpose of path analysis is to estimate the significance of hypothesized correlations among variables in a path model [23]. The study design of this research was based on Erikson’s theory, the ego development stages were influenced by the previous stages consequently, and the purpose of the research was to investigate the elderly adults with the eighth stage (integrity versus despair). Therefore, the eighth stage in Erikson’s theory was identified as the dependent variable and the first seven stages as independent variables. Theoretically, the ego development stages of Erikson`s theory were sequential and in order. One stage was influenced by the next stage, and the influences happened subsequently step by step through the eight stages. No stage would be exempted in the ego development process. The effect between one stage to the next stages was one-way relationship, which means the development of later stages could not influence the earlier stages. Based on these rules, the path model only showed the previous stage’s influence on the next stage. The “direct effect” was defined as the influence from one stage to the next stage, and the “indirect effect” was the influence from all previous stages except the previous stage.

From the literature, it is suggested that using the maximum likelihood (ML) method for estimation with sample size smaller than 100 was risky, while sample size over 500 would be adequate to use ML [24]. Because all of the data were continuous variables and the effective sample size in this study was only 292, the maximum likelihood method was exempted for estimation. The collected data were analyzed by using generalized least square (GLS) method [25]. The GLS estimator assumes the data fit normal distribution, and it is also characterized as asymptotically unbiased, consistent, efficient, and normally distributed full-information estimator [26]. To evaluate the model fit, various indices were checked in the results of path analysis; the goodness-of-fit index (GFI), adjusted goodness-of-fit index (AGFI), the root mean square error of approximation (RMSEA), and normed fit index (NFI) were used to assess the model fit. The results were: GFI = 0.92 and AGFI = 0.90; when these two values are greater than 0.90, it indicates good model fit. However, NFI = 0.87, which is a bit less than the required minimum of 0.9 [27,28]. The NFI = 0.87 indicated that the model improves the fit by 87% relative to the null model [29]. The RMSEA = 0.09 indicated acceptable fit [30]. The model fit could be accepted.

## 3. Results

The participants completed the questionnaire through an interview with the investigator. A total of 292 old older adults (65–90 years old) participated in the study, among whom 154 (52.7%) were women, and 138 (47.3%) were men. The mean age was 71 years (standard deviation = 6.2). In addition, of the participants, 52.1% were religious, 62.3% were retired, and only 9.9% lived alone. The path analysis model and results are shown in Figure 1.

The means for the eight stages of ego development ranged from 45.4 to 55.4, the highest scores located in ego integrity. A Pearson correlation analysis was performed by using SPSS (Statistical Product and Service Solutions) software version 22 (Table 1). All eight stages were significantly related to each other. The generativity (r = 0.77) and initiative (r = 0.32) stages were the most and least related to ego integrity, respectively.

The direct and indirect effects were estimated using the generalized least square (GLS) method procedure. The overall path analysis demonstrated that the previous stages exerted a significant direct effect on a given stage and indirect effects on later stages. According to Erikson’s theory, the ego development stages were sequential, and the previous stage influenced the later stage, and the later stages were all mediated by the previous stages (independent variable). Therefore, the influence from the seventh stage to the eighth stage was the only direct effect on the eighth stage. The other stages, except the seventh stage, influenced the eighth stage indirectly, and all of them impacted the eighth stage through the seventh stage (indirect effects).

The estimates of direct, indirect, and total effects on the eighth stage are shown in Table 2. The seventh stage showed the direct effect on the eighth stage (effect = 0.89). Comparing all the indirect effects of the previous six stages on the eighth stage, the largest effect was the indirect effect of the sixth stage on the eighth stage (indirect effect = 0.80). However, the sixth stage actually impacted the eighth stage with mediation by the seventh stage. It is worth noting that the seventh stage contributed a strong part (direct effect = 0.89) of the effects.

Next, stepwise regression analysis was used for evaluating the effects of each individual stage on ego integrity instead of the direct or indirect effects of previous stages. The ego integrity stage was recognized as the independent variable, and the previous seventh stages were defined as dependent variables. In the first step, the seventh stage was selected into the model, and other stages were excluded; the analysis of variance was statistically significant (F = 412.73, *p* < 0.01). In the second step, the fifth stage was added to the model, and the model also achieved statistical significance (F = 285.48, *p* < 0.01). Finally, the first stage was included in the regression model; the analysis of variance was statistically significant (F = 246.99, *p* < 0.01), and no variable could improve the model at statistical significance. In the stepwise regression analysis, the ego integrity stage was significantly predicted by the first trust stage (β = 0.42), fifth identity stage (β = 0.32), and seventh generativity stage (β = 0.48). The following equation was used for the prediction model (Table 3).
Stage 8 = (−9.16) + 0.48 × Stage 7 + 0.32 × Stage 5 + 0.42 × Stage 1

## 4. Discussion

Ego integrity versus despair (the eighth stage) is an important stage for older adults, particularly in being related to their life experiences in psychosocial development and facing death. This research aimed to explore the impacts of previous ego development stages on the elder stage (the eighth stage) with self-integrity. The results of this study revealed three conclusions. First, ego integrity is significantly influenced by the ego developments in the previous seven stages in Pearson’s correlation analysis, especially the adulthood (the seventh stage). Second, comparing to the other stages, the stages in adulthood (the seventh stage) had an irreplaceable and significant effect on ego integrity in path analysis results. Third, through regression analysis, the first, fifth, and seventh stages were able to significantly predict the ego integrity in the eighth stage.

In agreement with previous studies [11] and in line with Erickson’s theory, this study demonstrates that ego integrity is significantly related to the development of the previous seven stages, especially the seventh generativity stage. Besides, this study is the first study to apply path analysis for evaluating the direct and indirect effects of the previous seven stages on the ego integrity stage. The ego development in adulthood is the most influential on ego integrity in the elder stage. One possible explanation for this large influence of the adulthood stage is that adults are more capable of coping with suffering and challenges due to their resources and experiences. The previous research showed that by making sense of highly challenging life events, individuals develop more mature and complex approaches to thinking about themselves, others, and the world around them, which gradually and eventually become enduring aspects of their personality [15]. Individuals differ in how they respond to and perceive challenges, frustration, and uncertainty. As stated by the literature, “the factor that is important to consider in relation to ego development is how people respond to the impacts when difficult events occur [15]”. In addition, adults may have to take care of older adult relatives and raise the next generation, which may increase their awareness of life and death.

In general speaking, adults do not remember their infancy stage, but the research indicated that ego development is influenced by the individual’s trust hugely. Research indicated that mental development is important to later stages through the whole life, even in the elder stage [6,7,11]. In the regression model in this study, infancy (the first stage) remained a significant effect in the final regression model as a significant predictor; therefore, appropriate responses and caring feedback are critical in the infancy stage for establishing trust relationships in the long term.

In the adolescence period (the fifth stage), the individuals start to perform secondary sex characteristics; this stage is especially important in developing sex and role definitions [7]. Ego development in late adulthood could be traced back to psychosocial growth throughout the lifetime. A positive adolescent psychosocial development might ultimately influence ego development in old age [4]. The present study indicated that ego development in adolescence (the fifth stage) was one of the stages which could predict ego integrity, which echoed previous literature [4,11]. However, in Taiwan, most adolescents are under pressure of competition for school admission with their peers; the phenomenon is common in Asian culture as well. Under the huge stress of academic pressure in the eastern cultural context, ego development may be distorted and need further investigation.

This study has some limitations. First, this study was conducted in a cross-sectional design, without concerning time-related influences or causal inference. Second, the study findings are limited in generalizability to other cultures because all of the participants were from Taiwan. Further studies should investigate how ego development differs under various cultures, particularly between cultures in East Asia versus cultures in Europe and North America. In addition, further studies could consider using qualitative research design in ego development for exploring how the idiographic phenomenon differs between cultures in depth. Despite these limitations, this research provides evidence that the seventh stage, generativity, has the greatest influence on ego integrity. In general, individuals whose ego development has been perniciously affected by negative experiences must find resources to help themselves cultivate their mental well-being in the generativity stage so that they can face death and old age with more peace of mind.

According to the research results of the study, there are some suggestions to promote better ego development for individuals. This study showed that adulthood (the seventh stage) is the stage most correlated with ego development of older adults (the eighth stage); thus, it is suggested that individuals who have unresolved mental hurt or regret from early life stages should be treated before health care providers try to help the elderly person complete the ego integrity process. Infancy (the first stage), adolescence (the fifth stage), and adulthood (the seventh stage) have significantly impacted the elder stage in ego development with integrity mission; it is suggested that people should not only pay attention to adolescent and adulthood development but the infancy stage should also be noticed and cared for as well. For example, babies’ crying should not be neglected, despite the fact that those infancy experiences may not be remembered conscientiously in the future.

Furthermore, the development task of the adolescent is “identity versus role confusion” [7]. This study indicated that adolescence (the fifth stage) is one of the significant stages which could efficiently predict ego development in the older stage. In terms of caring for the adolescent, the priorities should include the concerns of their stress from academic tasks, from peer’s relationships, and gender as a role in society. In addition, adults or parents should avoid putting pressure on the adolescent. Instead of putting traditional values on the adolescents, offering supports to allow them to engage in things they are interested in and enthusiastic about may be helpful for better ego development. A previous study suggested that educators should support students to commit to a self-determined selection after careful deliberation [31]. Therefore, good communication and admitting to various points of view in the adolescent stage is important for more space in self-exploration.

No stage of Erikson’s Ego Development Theory could be neglected in successful ego development; in this era of an aging society, ego integrity becomes more and more important in dealing with elder mental issues or providing health care services.

## Figures and Tables

**Figure 1 ijerph-18-09490-f001:**
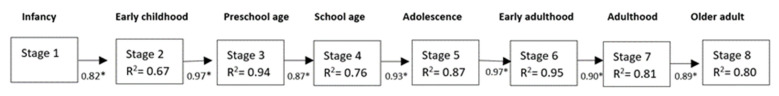
Path analysis model of Erickson’s theory of eight stages with direct and indirect effects (one-way sequential model and no stage could be omitted). Note: stage 1: trust versus mistrust, stage 2: autonomy versus shame, stage 3: initiative versus guilt, stage 4: industry versus inferiority, stage 5: identity versus role confusion, stage 6: intimacy versus isolation, stage 7: generativity versus stagnation, stage 8: integrity versus despair. *p* < 0.05 = *.

**Table 1 ijerph-18-09490-t001:** The relationship of the eight stages of ego development (*n* = 292).

	Range	Mean	SD	Skew	Kurtosis	Stage 1	Stage 2	Stage 3	Stage 4	Stage 5	Stage 6	Stage 7	Stage 8
Stage 1	34–64	52. 8	4.2	−0.834	2.760	1							
Stage 2	41–63	49.8	3.9	0.048	−0.230	0.51 **	1						
Stage 3	30–56	45.4	4.5	−0.203	0.290	0.41 **	0.41 **	1					
Stage 4	36–64	52.6	5.1	−0.438	0.501	0.59 **	0.57 **	0.43 **	1				
Stage 5	35–63	49.0	4.6	0.366	0.571	0.37 **	0.40 **	0.25 **	0.56 **	1			
Stage 6	39–62	50.8	4.9	−0.189	−0.387	0.55 **	0.40 **	0.47 **	0.40 **	0.38 **	1		
Stage 7	39–67	55.4	4.6	−0.250	0.566	0.70 **	0.60 **	0.39 **	0.70 **	0.44 **	0.51 **	1	
Stage 8	38–69	55.4	4.2	−0.138	0.479	0.72 **	0.56 **	0.32 **	0.67 **	0.58 **	0.49 **	0.77 **	1

Note: stage 1: trust versus mistrust, stage 2: autonomy versus shame, stage 3: initiative versus guilt, stage 4: industry versus inferiority, stage 5: identity versus role confusion, stage 6: intimacy versus isolation, stage 7: generativity versus stagnation, stage 8: integrity versus despair. ** *p* < 0.01.

**Table 2 ijerph-18-09490-t002:** The effects from previous seven stages to the eighth stage (*n* = 292).

Variables		Effects
	Direct	Indirect
Stage 1		0.52
Stage 2		0.63
Stage 3		0.65
Stage 4		0.74
Stage 5		0.79
Stage 6		0.80
Stage 7	0.89	

Note: stage 1: trust versus mistrust, stage 2: autonomy versus shame, stage 3: initiative versus guilt, stage 4: industry versus inferiority, stage 5: identity versus role confusion, stage 6: intimacy versus isolation, stage 7: generativity versus stagnation, stage 8: integrity versus despair.

**Table 3 ijerph-18-09490-t003:** The stepwise regression model in the first seven stages to predict ego integrity (*n* = 292).

Adjusted R^2^	F	Model	β	Unstandardized Coefficients Std. Error	Standardized Coefficients Beta	T	*p* Value	95% CI for B
0.72	246.99 **	(Constant)	−9.16	2.40		−3.82	0.00	(−13.89, −4.434)
		Stage 7	0.48	0.05	0.41	9.20	0.00	(0.38, 0.58)
		Stage 5	0.32	0.04	0.28	8.01	0.00	(0.24, 0.40)
		Stage 1	0.42	0.06	0.33	7.60	0.00	(0.31, 0.53)

Note: stage 1: trust versus mistrust, stage 5: identity versus role confusion, stage 7: generativity versus stagnation. ** *p* < 0.01.

## Data Availability

Data is contained within the article. The data presented in this study are available in the article.

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
