# Peer review of "Predicting Ego Integrity Using Prior Ego Development Stages for Older Adults in the Community"

_ijerph, 2021, doi:10.3390/ijerph18189490_

Round 1
Reviewer 1 Report
table 1: the design of table 1 suggests that stages are correlate among each other and not to ego integrity score, also results of ego integrity scores are missing
table 2 and Figure 1 show the same results, please exclude one of both
table 3: please provide results by B, SE, 95%CI for B and p value
line 177: reference is missing
line 193: result of 1.54 is not presented in table 2
Author Response
Dear Reviewers:
Thanks very much for taking your time to review this manuscript. I really appreciate all your comments and suggestions! Please find my itemized responses in below and my revisions in the re-submitted files.

Reviewer 2 Report
I found the manuscript interesting. It presents the eight stages of psychosocial development of older adults in Taiwan. Nevertheless, I still have some major problems with the manuscript, and I will summarize the problems and comments below.
- The quantitative analysis part of the manuscript needs major revision. Total effect, direct effect and indirect effect are clearly defined in path analysis, but the definitions given by the authors in this manuscript do not completely conform to their conventional definitions, so more standardized path analysis should be used.
- The contents described in lines 186-195 are inconsistent with the values in Table 2 or Figure 1.
- The path analysis model in Figure 1 does not show all the influencing paths described by the authors. Except the one-way influence of the adjacent previous stage on the next stage, no other path are shown in the figure, so this figure can not correspond to the author's text description.
- It is mentioned in lines 187-188 that the unweighted least square method is used for estimation. However, these eight stages are highly correlated, so is this method appropriate? The reliability and robustness of the estimation results are questionable.
- Table 3 shows the final results of stepwise regression analysis, but it does not provide the screening conditions and process of stepwise regression.
- The equation mentioned in line 204 does not appear in the manuscript.
- The relevant contents of the questionnaire survey are not clearly written, and the revised version needs to supplement the sampling methods and interview details.
- The discussion on the practical significance of data analysis results should be strengthened in the revised edition.
Author Response

(The authors gave the same response as above.)

Reviewer 3 Report
This study examined the prediction of ego integrity in old age in relation to the seven preceding ego developmental stages in community-aged adults. The results suggest that the ego developmental stages of adulthood, especially the seventh stage of generativity, have the greatest impact on old age. In modern society, where the elderly population is increasing, this study provides important issues for mental development that will enable people to face death and old age with peace of mind.
However, I think that the following points about the publication need to be considered.
(1) In the description of the literature, although there are four authors in reference number 19, only the first author is listed. This needs to be corrected so that it is in line with the submission rules.
(2) In the main text, reference number 14 is cited in paragraphs 223 and 227 of the discussion section, but only two authors are listed. Similarly, in the discussion section, reference number 19 is cited in paragraph 232, but there are only two authors listed. In the same way, Reference 19 has four authors, and we believe that four authors should be listed in accordance with the previously published articles in this journal. In addition, we believe that it is necessary to confirm the authors' statements when citing the literature, in accordance with the submission rules.
Author Response

(The authors gave the same response as above.)

Round 2
Reviewer 2 Report
The manuscript has made great progress after being revised by the authors, but there are still some minor deficiencies.
The quality of quantitative analysis needs to be improved, especially, the result of path analysis is still not clear enough. The model in Figure 1 is too simple to be called a path analysis model accurately. The path analysis model should include all possible direct and indirect influence paths. The authors should further revise Figure 1, or at least change its name.
In addition, the current revision of Figure 1 is too rough to meet the requirements of paper publishing. It needs decoration to make it more refined.
Author Response
Thanks for your letter dated 5.Sep. 2021. We have already revised the manuscript and uploaded it via the submitted system today. We thank you for the time and their suggestions have enabled us to improve our manuscript. In the revised version we have made modifications as suggested by reviewers. Our responses are given directly afterward in a different color.
